# The Combined Conditions of Photoperiod, Light Intensity, and Air Temperature Control the Growth and Development of Tomato and Red Pepper Seedlings in a Closed Transplant Production System

**Hyunseung Hwang** [1,†] (ID), **Sewoong An** [2,†] (ID), **Minh Duy Pham** [1,3] (ID), **Meiyan Cui** [1,3] (ID) **and Changhoo Chun** [1,3,*]

1   Department of Agriculture, Forestry and Bioresources, Seoul National University, Seoul 08826, Korea; behong47@snu.ac.kr (H.H.); minhduy512@yahoo.com (M.D.P.); cuimeiyan@snu.ac.kr (M.C.)
2   Department of Horticultural Crop Research, National Institute of Horticultural and Herbal Sciences, Rural Development Administration, Wanju 55365, Korea; woong0911@korea.kr
3   Research Institute of Agriculture and Life Sciences, Seoul National University, Seoul 08826, Korea
*   Correspondence: changhoo@snu.ac.kr; Tel.: +82-2-880-4567
†   These authors contribute equally to this work.

**Abstract:** Understanding environmental factors is essential to maximizing the biomass production of plants. There have been many studies on the effects of the photosynthetic photon flux (PPF), photoperiod and air temperature as separate factors affecting plants, including under a closed transplant production system (CTPS). However, few studies have investigated the combined effects of these factors on plant growth. Germinated tomato and red pepper seedlings were transferred to three different photoperiods with five different photosynthetic photon fluxes (PPFs) at an air temperature of 25/20 °C to investigate plant growth under a different daily light integral (DLI). Three different air temperatures, 23/20, 25/20, and 27/20 °C (photo/dark periods), with five different PPFs were used to examine plant growth under different DIFs (difference between the day and night temperature). Increasing the DLI from 4.32 to 21.60 mol·m$^{-2}$·d$^{-1}$, either by increasing the photoperiod or PPF, improved the growth of seedlings in both cultivars. However, when comparing treatments that provided the same DLI, tomato seedlings had s significantly higher growth when grown under longer photoperiods and s lower PPF. Even in higher DLI conditions, reduced growth due to higher PPF indicated that excessive light energy was a limiting factor. At 23 and 25 °C, tomato seedlings showed similar correlation curves between growth and PPF. However, at the higher temperature of 27 °C, while the slope of the curve at low PPFs was similar to that of the curves at lower temperatures, the slope at high PPFs was flatter. On the other hand, red pepper seedlings displayed the same correlation curve between growth and PPF at all tested temperatures, and red pepper plants accumulated more dry weight even at higher temperatures. These results suggested that the combination effect was more useful to observe these overall tendencies, especially in reacting to a second factor. This will provide us with more information and a deeper understanding of plant characteristics and how they will behave under changing environments.

**Keywords:** air temperature; light intensity; photoperiod; closed transplant production system

## 1. Introduction

Recently, plant factories have become more popular in the cultivation and production of vegetable seedlings [1], herbs [2], and medicinal plants [3,4] due to their numerous advantages over conventional

protected horticulture. Especially in transplant production, closed systems using artificial lighting, which are called closed transplant production systems (CTPSs), have been designed and adopted for the cultivation and propagation of seedlings and transplants [5]. These systems can reduce labor and resource consumption [6], increase the production curve, and improve plant quality due to the ability to control environmental factors for optimal growth conditions without being affected by outside weather [5].

However, to successfully apply CTPS in transplant production it is necessary to study the responses of plants to environmental factors for the selection of suitable growth conditions. Among the factors that affect plants in CTPS, air temperature, photosynthetic photon flux (PPF), and photoperiod are the most important ones. PPF and photoperiod influence photosynthesis and carbohydrate gain, thereby determining the growth rate and vigor [7–9]. On the other hand, air temperature affects transpiration, morphology, and photosynthetic efficiency [10–12]. Together, these factors have a great impact on plant growth and quality. Therefore, it is essential to understand how plants respond to changes in these factors for successful CTPS operation.

There have been many studies on the effects of PPF, photoperiod, and air temperature as separate factors affecting plant growth in closed systems such as CTPS. However, few studies have investigated the combined effects of these factors on plant growth. PPF, photoperiod, and air temperature in particular do not affect plants independently. Rather, their influences on physiological processes such as photosynthesis, transpiration, and morphological development are interconnected [13]. If one of the environmental factors changes, the results may vary significantly depending on the rest of the environmental factors. Therefore, it is important to examine the complex interactions of various environmental factors.

Furthermore, most studies focus only on the differences among specific treatment conditions instead of investigating the trend in the plant responses to environmental factors as a whole [14]. As each plant species has unique nature, it is more useful to observe these overall tendencies, especially when the plant is reacting to a second factor. This will provide us with more information and a deeper understanding of plant characteristics and how they will behave under changing environments. This knowledge not only allows us to adjust and optimize the conditions inside a CTPS for maximum plant growth and desirable quality but can also be applied in the field for the adaptation of the cultivation to the changing weather.

Therefore, in this study we evaluated the combined effects of air temperature, light intensity, and photoperiod on the growth and development of tomato and red pepper seedlings. By using regression analyses and careful comparisons, we attempted to uncover the dynamic interactions among these factors and gain a better understanding of the natures and preferences of tomato and red pepper seedlings in terms of environmental conditions. From these data, we will be able to set the optimum air temperature, light intensity, and photoperiod in CTPS for each species to serve various purposes.

## 2. Materials and Methods

### 2.1. Plant Materials and Growth Conditions

On 1 February, seeds of tomato (*Solanum lycopersicum* L. cv. Dotaerangdia, Takii, Seoul, Korea) and red pepper (*Capsicum annuum* L. cv. Shinhong, Nongwoo Bio) were sown on a 162-cell plug tray (2.5 × 2.5 cm; 15 mL volume) filled with commercial growing medium (Biomedia, Hungnong Seed Co., Ansung, Korea) and cultivated in a closed system. All the treatments used white LED lamps (model v3.0W, Future Green Co., Ltd., Gyeonggi, Korea, Figure 1). The seedlings were grown at a $CO_2$ concentration of 400 $\mu mol \cdot mol^{-1}$ and a relative humidity of 70–90%. Seedlings were subirrigated every 2 or 3 days with pH 6.5 and EC 1.4 $dS \cdot m^{-1}$ of nutrient solution. The pH and EC (electrical conductivity) of the nutrient solution were measured using a digital pH meter and an EC meter (Horiba D-54, Tokyo, Japan).

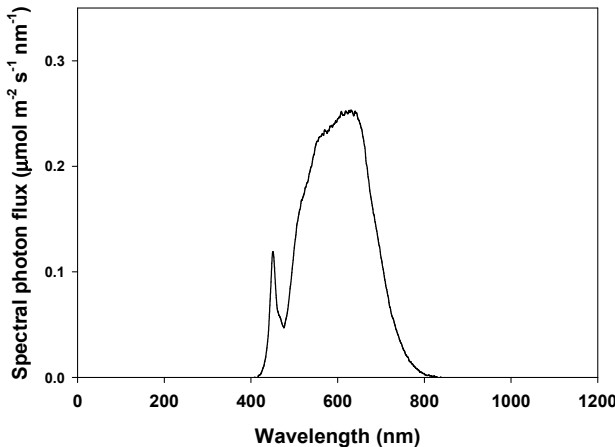

**Figure 1.** The spectral distribution of white LEDs used in this experiment. Spectral scans were conducted at the center of the tray with a light intensity of 100 $\mu mol \cdot m^{-2} \cdot s^{-1}$.

## 2.2. Light Treatment_Light Intensity and Photoperiod

The experiment consisted of 15 treatments, with 7 plants × 3 replicates per treatment. The germinated seedlings were transferred to treatment conditions, and the seedlings were grown under three different photoperiods (Table 1) with five different PPFs (Table 2) at an air temperature of 25/20 °C for 15 days and 23 days for tomato and red pepper, respectively.

**Table 1.** Daily light integral (DLI) of this experiment.

| Photoperiod (h·d$^{-1}$) | PPF ($\mu mol \cdot m^{-2} \cdot s^{-1}$) | DLI (mol·m$^{-2} \cdot d^{-1}$) |
|---|---|---|
| 12 | 100 | 4.32 |
| | 150 | 6.48 |
| | 200 | 8.64 |
| | 250 | 10.80 |
| | 300 | 12.96 |
| 16 | 100 | 5.76 |
| | 150 | 8.64 |
| | 200 | 11.52 |
| | 250 | 14.40 |
| | 300 | 17.28 |
| 20 | 100 | 7.20 |
| | 150 | 10.80 |
| | 200 | 14.40 |
| | 250 | 18.00 |
| | 300 | 21.60 |

**Table 2.** Light intensity treatment of this experiment.

| Treatment | Photosynthetic Photon Flux ($\mu mol \cdot m^{-2} \cdot s^{-1}$) | | | | Total |
|---|---|---|---|---|---|
| | 400–500 | 500–600 | 600–700 | 700–800 | |
| PPF100 | 9.1 ± 0.5 | 38.6 ± 2.1 | 40.5 ± 2.2 | 7.3 ± 0.4 | 95.4 ± 5.1 |
| PPF150 | 14.0 ± 0.5 | 59.6 ± 2.3 | 62.4 ± 2.4 | 11.2 ± 0.4 | 147.1 ± 5.7 |
| PPF200 | 18.5 ± 0.8 | 79.1 ± 3.3 | 82.9 ± 3.5 | 14.9 ± 0.6 | 195.4 ± 8.2 |
| PPF250 | 23.4 ± 0.9 | 99.6 ± 3.7 | 104.3 ± 3.9 | 18.7 ± 0.7 | 246.0 ± 9.2 |
| PPF300 | 28.0 ± 1.4 | 119.6 ± 5.9 | 125.3 ± 6.2 | 22.5 ± 1.1 | 295.5 ± 14.6 |

### 2.3. Air Temperature Treatment

The experiment consisted of 15 treatments, with 7 plants × 3 replicates per treatment. The germinated seedlings were transferred to treatment conditions consisting of three different air temperatures, 23/20, 25/20, and 27/20 °C (photo/dark periods) (Figure 2), with five different photosynthetic photon fluxes (Table 3) with a 20 h photoperiod.

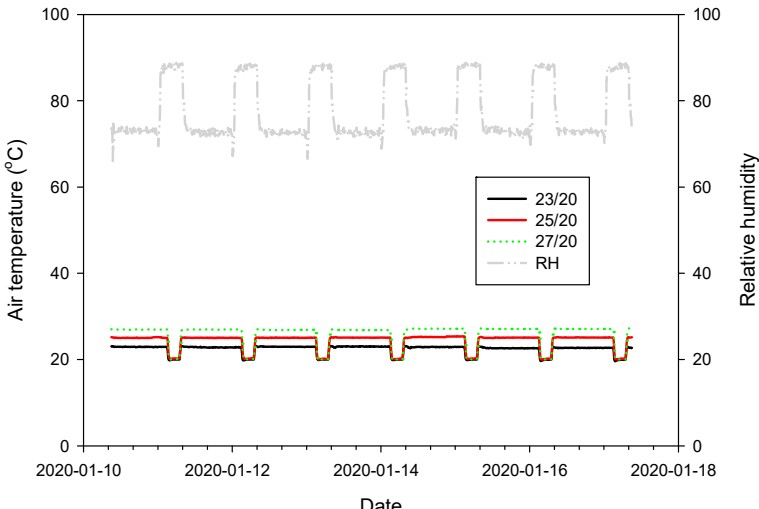

**Figure 2.** Air temperature and relative humidity at 20 h·d$^{-1}$.

### 2.4. Plant Growth Characteristics

The seedlings were harvested for the evaluation of growth parameters, including hypocotyl length, stem diameter, leaf number, leaf area, fresh weight, and dry weight of tomato and red pepper seedlings, which were measured at 17 and 25 days after sowing (DAS). Stem diameter was measured using a Vernier caliper (CD-20CPX, Mitutoyo Co., Japan). Leaf area was measured using leaf area meters (Li-3100; LI-COR, Lincoln, NE, USA). Dry weight was measured after drying at 80 °C for 3 days. The leaf area index (LAI), dry matter content (DMC), compactness, and light use efficiency (LUE) were calculated using the following formula:

$$\text{Leaf area index} = \text{leaf area (cm}^2)/(\text{plug tray area (58} \times \text{24 cm}^2)/162);$$

$$\text{Dry matter content} = (\text{dry weight/shoot fresh weight}) \times 100;$$

$$\text{Compactness} = \text{shoot dry weight (g)/hypocotyl length (cm);}$$

$$\text{Light use efficiency} = \text{shoot dry weight (g)/DLI (mol·m}^{-2}\text{·d}^{-1}).$$

### 2.5. Statistical Analysis

The experimental data were analyzed by the Statistical Analysis System (SAS) for Window version 9.4 (SAS Institute Inc., Cary, NC, USA) using a two-way factorial analysis of variance (ANOVA). Significant differences were considered at $p < 0.05$, 0.01, and 0.001.

## 3. Results

### 3.1. Effects of Different Light Intensities and Photoperiods on Tomato and Red Pepper Seedlings

Figure 3 shows representative images of tomato seedlings at 17 days after sowing (DAS) and red pepper seedlings at 25 DAS under different light intensities and photoperiods.

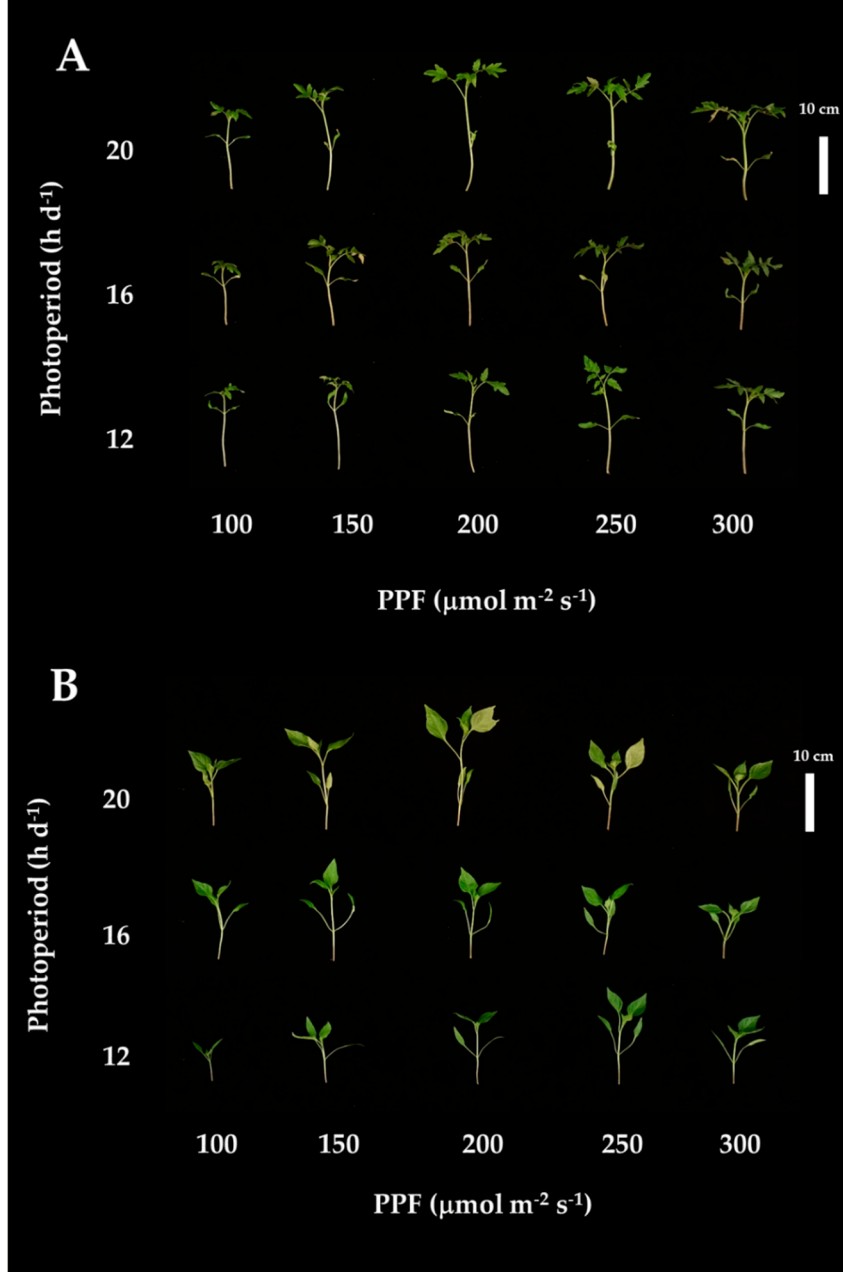

**Figure 3.** Seedlings of tomato (**A**) at 17 days after sowing (DAS) and red pepper (**B**) at 25 DAS as influenced by the different light intensities and photoperiods.

### 3.1.1. Shoot Length and Stem Diameter

Increasing the PPF from 100 to 300 $\mu mol \cdot m^{-2} \cdot s^{-1}$ influenced the shoot length and stem diameter of tomato and red pepper seedlings differently depending on the photoperiod (Figure 4). At photoperiods of 12 and 20 $h \cdot d^{-1}$, the shoot length of tomato was the highest at 200 and 150 $\mu mol \cdot m^{-2} \cdot s^{-1}$, respectively. Further increases in the PPF reduced the shoot length. In the case of red pepper, at 12 and 16 $h \cdot d^{-1}$ photoperiods, the shoot length was the highest at 200 $\mu mol \cdot m^{-2} \cdot s^{-1}$ and decreased with higher PPFs. At 20 $h \cdot d^{-1}$, there was no difference in the shoot length of red pepper among PPFs of 100, 150, and 200 $\mu mol \cdot m^{-2} \cdot s^{-1}$, while a higher PPF reduced the shoot length.

An increase in PPF from 100 to 250 $\mu mol \cdot m^{-2} \cdot s^{-1}$ promoted the stem diameter of tomato seedlings among all the tested photoperiods, but there was no significant difference between 250 and 300 $\mu mol \cdot m^{-2} \cdot s^{-1}$. The stem diameter of red pepper was the highest at 200 $\mu mol \cdot m^{-2} \cdot s^{-1}$ regardless of

the photoperiod, while those at 12 and 20 h·d⁻¹ had no significant differences if the PPF was higher than 200 and 150 μmol·m⁻²·s⁻¹, respectively.

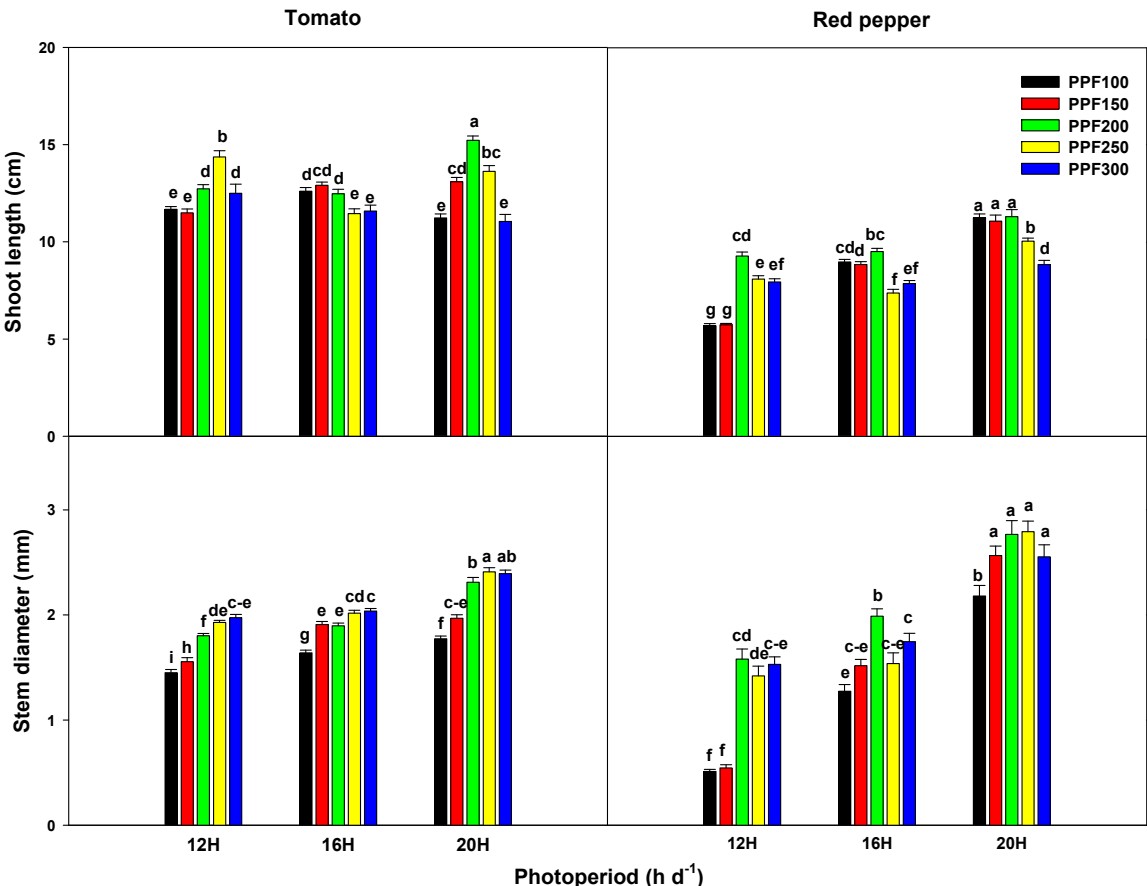

**Figure 4.** Shoot length and stem diameter of tomato at 17 days after sowing (DAS) and red pepper at 25 DAS, as influenced by the different light intensities and photoperiods (n = 7, r = 3). Different letters indicate significant differences by Duncan's multiple range test at *p* = 0.05.

### 3.1.2. Leaf Characteristics

Leaf number, leaf area, and leaf area index (LAI) in tomato all showed general improvements when the photoperiod increased from 12 to 20 h·d⁻¹ under the same PPF (Figure 5). Conversely, an increase in the PPF from 100 to 250 μmol·m⁻²·s⁻¹ under the same photoperiod also promoted these parameters. However, when the PPF increased to 300 μmol·m⁻²·s⁻¹, these parameters slightly decreased compared to treatments with a PPF of 250 μmol·m⁻²·s⁻¹ under the same photoperiod, although significant differences were only observed in the leaf area and LAI at a photoperiod of 20 h·d⁻¹. Overall, treatments 20/200 and 20/250 were the highest among all treatments in terms of these parameters. In the case of red pepper, there was a difference between low and high PPFs in the responses of these parameters to the increase in photoperiod. When the PPF was below 250 μmol·m⁻²·s⁻¹, all three parameters showed significant increases when the photoperiod increased from 12 to 16 to 20 h·d−1. However, at PPFs of 250 and 300 μmol·m⁻²·s⁻¹, while a photoperiod of 20 h·d⁻¹ still gave the highest results among all photoperiods under the same PPF, there was no significant difference in these parameters between photoperiods of 12 and 16 h·d⁻¹. On the other hand, when comparing among treatments within the same photoperiod, the leaf number generally increased with the PPF, although there was no significant difference among treatments with a PPF higher than 200 μmol·m⁻²·s⁻¹. Conversely, the leaf area and LAI showed a similar increase-decrease pattern to tomato. Nevertheless, among all treatments the highest results were obtained with PPFs of 150 to 300 μmol·m⁻²·s⁻¹ at the same photoperiod of 20 h·d⁻¹.

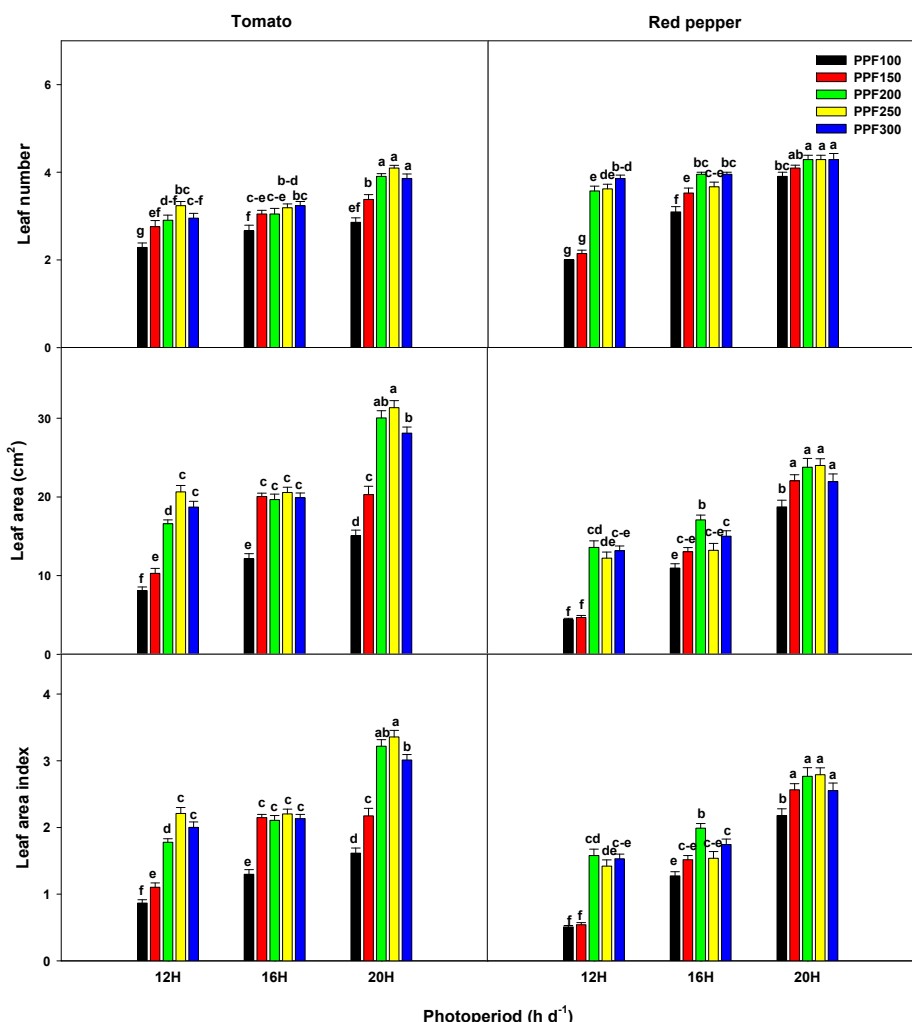

**Figure 5.** Leaf number, leaf area, and leaf area index (LAI) of tomato at 17 days after sowing (DAS) and red pepper at 25 DAS, as influenced by the different light intensities and photoperiods (n = 7, r = 3). Different letters indicate significant differences by Duncan's multiple range test at *p* = 0.05.

### 3.1.3. Dry Weight, Dry Matter Content, and Compactness

Increasing the DLI from 4.32 to 21.60 mol·m$^{-2}$·d$^{-1}$, either by increasing the photoperiod or PPF, improved the dry weight (DW), dry matter content (DMC), and compactness of seedlings in both cultivars. For instance, at a PPF of 200 μmol·m$^{-2}$·s$^{-1}$, increasing the photoperiod from 12 to 20 h·d$^{-1}$ increased the DW of tomato and red pepper seedlings by 143% and 181%, the DMC by 21% and 20%, and the compactness by 104% and 111%, respectively. Under the same photoperiod, the DW, DMD, and compactness increased linearly with PPF. Tomato and red pepper seedlings grown under a 20 h·d$^{-1}$ photoperiod showed increases in DW by 121% and 212%, in DMC by 24% and 5%, and in compactness by 143% and 171% as the PPF increased from 50 to 250 μmol·m$^{-2}$·s$^{-1}$. Similarly, under a shorter photoperiod of 16 h·d$^{-1}$, increasing the PPF increased the DW by 183% and 210%, the DMC by 24% and 15%, and the compactness by 194% and 141% for tomato and red pepper seedlings, respectively. Overall, the DW, DMC, and compactness showed a linear correlation with DLI (Figure 6).

However, it is notable that, in a comparison between treatments that provided the same DLI (treatments 12/200 and 16/150; treatments 12/250 and 20/150; treatments 16/250 and 20/200), tomato seedlings had significantly higher DWs when grown under longer photoperiods and lower PPFs. In the case of red pepper, under the same DLI of 8.64 mol·m$^{-2}$·d$^{-1}$ no difference in DW was found between the two light settings of treatments 12/200 and 16/150. However, at DLIs of 10.80 and 14.40 mol·m$^{-2}$·d$^{-1}$, a similar improvement in the DW of red pepper seedlings under longer photoperiods could be observed.

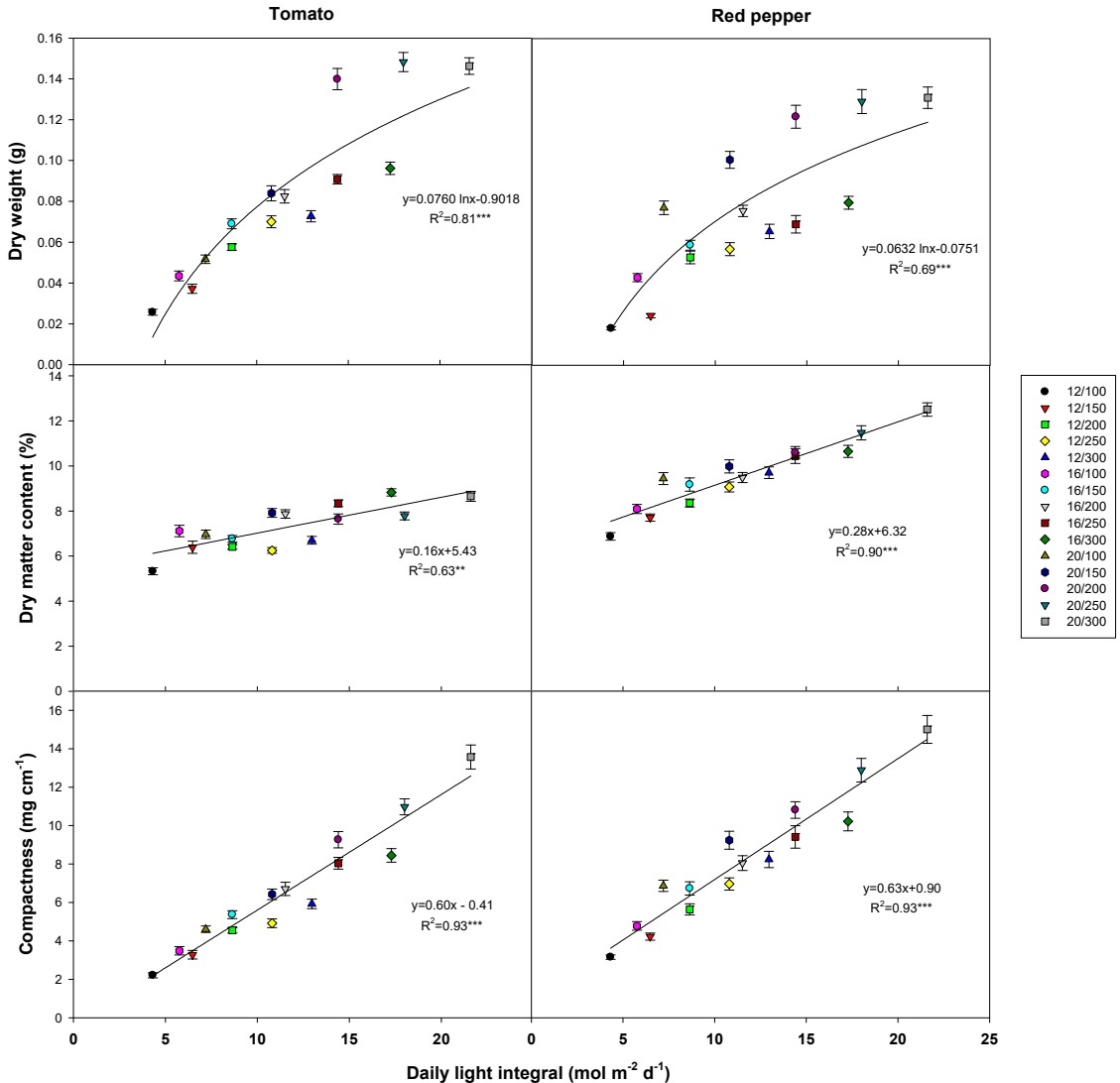

**Figure 6.** Regression analyses of the dry weight, dry matter content, and compactness related to light intensity and photoperiod. The value of each point is the average dry weight, dry matter content, and compactness of tomato at 17 days after sowing (DAS) and red pepper at 25 DAS, as influenced by the light intensity and photoperiod. Bars refer to the standard error.

Likewise, tomato seedlings grown under 150 $\mu$mol$\cdot$m$^{-2}\cdot$s$^{-1}$ for 20 h$\cdot$d$^{-1}$ and 250 $\mu$mol$\cdot$m$^{-2}\cdot$s$^{-1}$ for 16 h$\cdot$d$^{-1}$ had significantly greater DMCs than those grown under 250 $\mu$mol$\cdot$m$^{-2}\cdot$s$^{-1}$ for 10 h$\cdot$d$^{-1}$ and 200 $\mu$mol$\cdot$m$^{-2}\cdot$s$^{-1}$ for 20 h$\cdot$d$^{-1}$ with the same DLI of 10.80 and 14.40 mol$\cdot$m$^{-2}\cdot$d$^{-1}$, respectively, although those grown under DLI of 8.64 mol$\cdot$m$^{-2}\cdot$d$^{-1}$ showed no difference in DMC. In the case of red pepper, DMC showed no difference between the two light settings with the same DLI of 14.40 mol$\cdot$m$^{-2}\cdot$d$^{-1}$. However, at DLIs of 8.64 and 10.80 mol$\cdot$m$^{-2}\cdot$d$^{-1}$, the DMCs of red pepper grown under 150 $\mu$mol$\cdot$m$^{-2}\cdot$s$^{-1}$ for 16 h$\cdot$d$^{-1}$ and 150 $\mu$mol$\cdot$m$^{-2}\cdot$s$^{-1}$ for 20 h$\cdot$d$^{-1}$ were significantly greater than those grown under higher PPFs and shorter photoperiods.

A similar trend in the compactness of tomato and red pepper seedlings was observed. Under the same DLIs of 8.64, 10.80, and 14.40 mol$\cdot$m$^{-2}\cdot$d$^{-1}$, seedlings grown under 150 $\mu$mol$\cdot$m$^{-2}\cdot$s$^{-1}$ for 16 h$\cdot$d$^{-1}$, 150 $\mu$mol$\cdot$m$^{-2}\cdot$s$^{-1}$ for 20 h$\cdot$d$^{-1}$, and 200 $\mu$mol$\cdot$m$^{-2}\cdot$s$^{-1}$ for 20 h$\cdot$d$^{-1}$ had significantly greater compactness than those grown under higher PPFs and shorter photoperiods.

### 3.2. Effects of Different Air Temperatures (Photo and Dark Periods) and Light Intensities on Tomato and Red Pepper Seedlings

Figure 7 shows representative images of tomato seedlings at 17 days after sowing (DAS) and red pepper seedlings at 25 DAS under different light intensities and photoperiods.

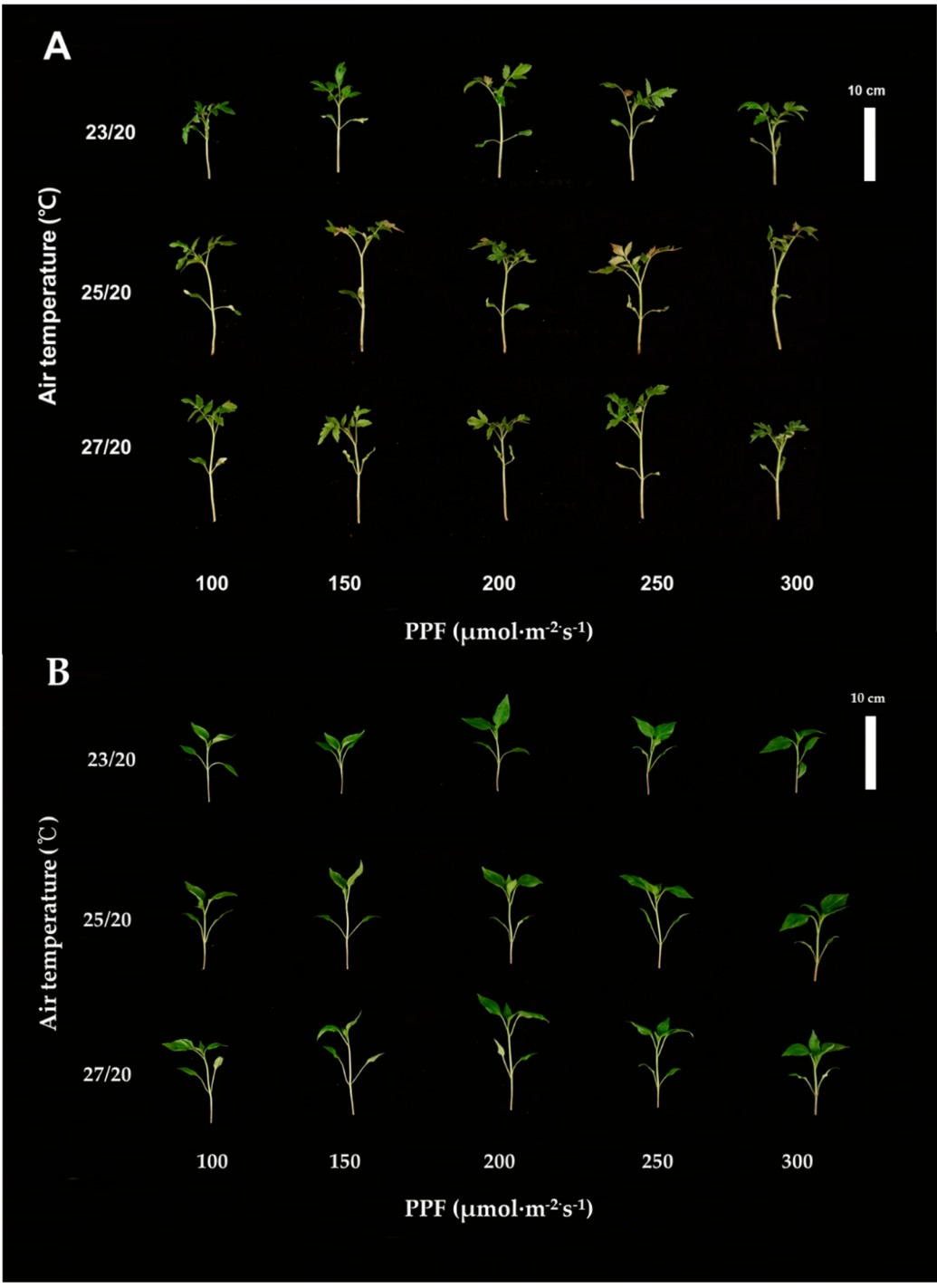

**Figure 7.** Seedlings of tomato (**A**) at 17 days after sowing (DAS) and red pepper (**B**) at 25 DAS, as influenced by the different light intensities and air temperatures of the photo- and dark periods.

### 3.2.1. Shoot Length and Stem Diameter

The response to changes in the PPFs of tomato and red pepper in terms of shoot length differed significantly depending on the photo/dark period air temperature (Figure 8). At 23/20 °C, tomato and red pepper generally showed similar patterns, with shoot length increasing as the PPF increased up to 200 $\mu mol \cdot m^{-2} \cdot s^{-1}$ in tomato and 250 $\mu mol \cdot m^{-2} \cdot s^{-1}$ in red pepper, and then decreasing as the PPF increased further to 300 $\mu mol \cdot m^{-2} \cdot s^{-1}$. At 25/20 and 27/20 °C, however, tomato displayed an opposite trend, with the shoot length generally reduced as PPF increased, to the lowest at 250 $\mu mol \cdot m^{-2} \cdot s^{-1}$ under 25/20 °C and at 200 $\mu mol \cdot m^{-2} \cdot s^{-1}$ under 27/20 °C, then increased as the PPF increased further. In the case of red pepper, there was no significant difference among the PPFs from 100 to 200 $\mu mol \cdot m^{-2} \cdot s^{-1}$ at these temperatures, but the shoot length slightly decreased as the PPF increased further to 250 and $\mu mol \cdot m^{-2} \cdot s^{-1}$. Nevertheless, the response of shoot length to an increase in temperature at the same PPF was consistent in each species. For tomato, the shoot length at the same PPF was higher at 25/20 °C than at 23/20 °C, but there was no further significant improvement as the temperature increased to 27/20 °C. At least in the case of red pepper seedlings, on the other hand, they still showed a significant increase in shoot length as the photoperiod temperature increased up to 27 °C in all PPFs.

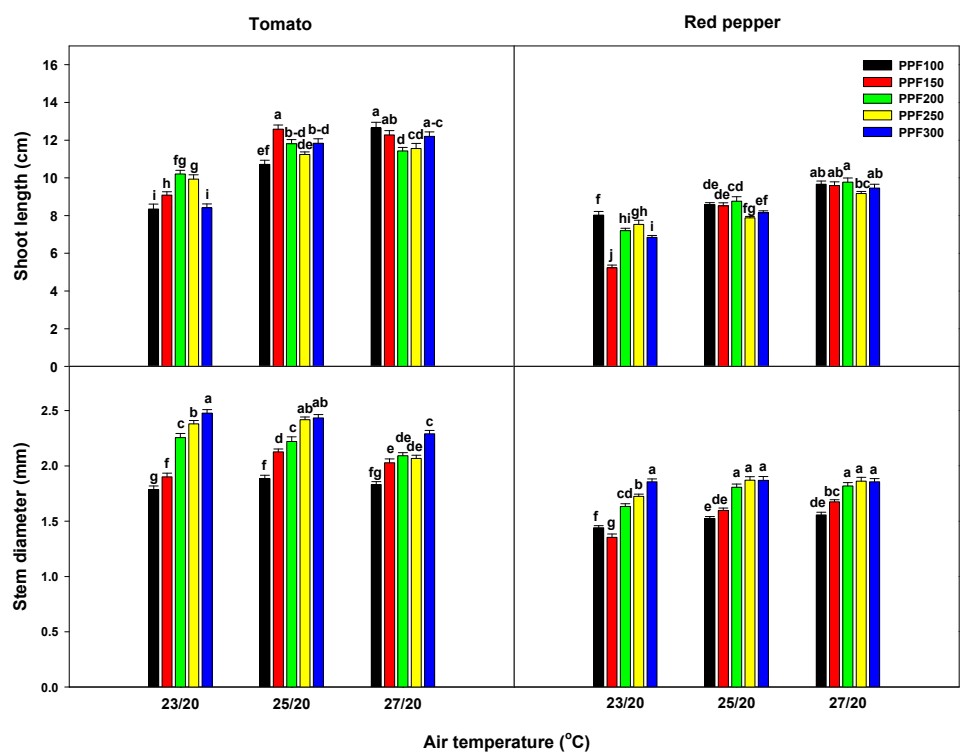

**Figure 8.** Shoot length and stem diameter of tomato at 17 days after sowing (DAS) and red pepper at 25 DAS, as influenced by the different light intensities and air temperatures of the photo- and dark periods (n = 7, r = 3). Different letters indicate significant differences by Duncan's multiple range test at *p* = 0.05.

The response of stem diameter to changes in PPF and temperature was more straightforward. In both species, the stem diameter showed an increasing trend as the PPF increased at the same temperature, although there was no significant difference in red pepper among PPFs higher than 150 $\mu mol \cdot m^{-2} \cdot s^{-1}$ at 25 and 27 °C. Across different temperatures at the same PPF level, however, the stem diameter in tomato showed a slight decrease as the temperature increased from 25 to 27 °C, while that of red pepper was not significantly different between 25 and 27 °C, except at 150 $\mu mol \cdot m^{-2} \cdot s^{-1}$.

### 3.2.2. Leaf Characteristics

In tomato, while there was no clear pattern in leaf number in response to changes in temperature and PPF, leaf area and LAI displayed a clear upward trend as PPF increased under all temperature settings. When comparing temperatures under the same PPF, however, there was a decrease in leaf number as the temperature increased from 25 to 27 °C, except at a PPF of 300 $\mu$mol·m$^{-2}$·s$^{-1}$. In red pepper, leaf number showed an overall increase as the PPF increased under all temperature settings. This pattern was only observed in leaf area and LAI at a temperature of 23/20 °C. In the other two temperature settings, no significant improvement was observed as the PPF increased beyond 200 $\mu$mol·m$^{-2}$·s$^{-1}$ (Figure 9).

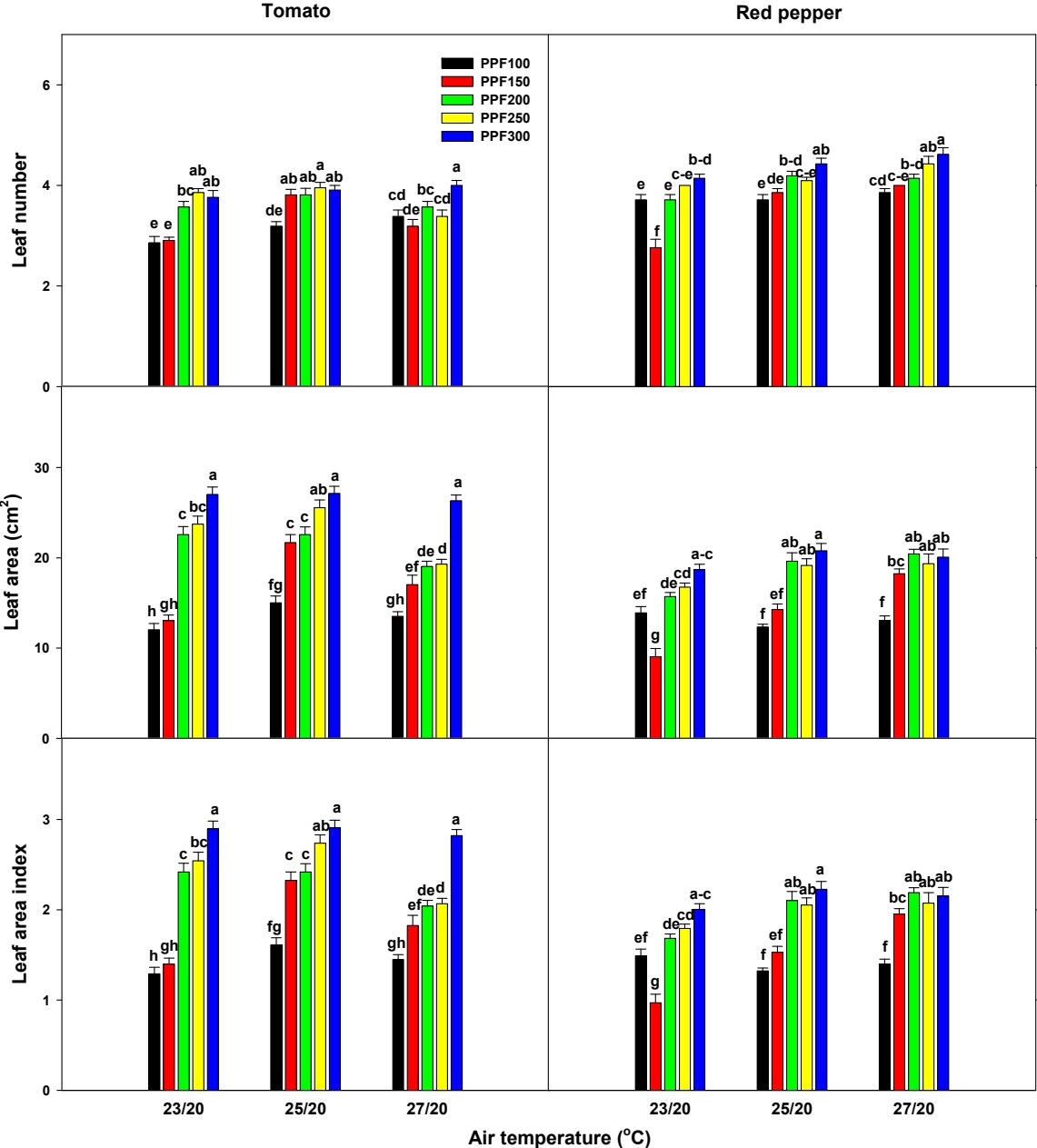

**Figure 9.** Leaf number, leaf area, and leaf area index (LAI) of tomato at 17 days after sowing (DAS) and red pepper at 25 DAS, as influenced by the different light intensities and air temperatures of the photo- and dark periods (n = 7, r = 3). Different letters indicate significant differences by Duncan's multiple range test at *p* = 0.05.

### 3.2.3. Dry Weight, Dry Matter Content, and Compactness

The DWs of both tomato and red pepper showed a strong positive logarithmic correlation with PPF (coefficients of determination were 0.79 and above) (Figure 10). An increase in PPF led to a higher DW in both species, but at higher PPFs this effect was less pronounced than at low PPFs. More importantly, the models revealed the differences between tomato and red pepper in the response to changes in PPF under different temperatures. At photoperiod temperatures of 23 and 25 °C, tomato showed similar correlation curves between DW and PPF. However, at the higher temperature of 27 °C, while the slope of the curve at low PPFs was similar to that of the curves at lower temperatures, the slope at high PPFs was flatter. On the other hand, red pepper displayed the same correlation curve between DW and PPF at all tested temperatures, albeit the curve at 27 °C was slightly higher at altitude than that at 25 °C, which in turn was higher than that at 23 °C, showing that red pepper plants accumulated more DW at higher temperatures.

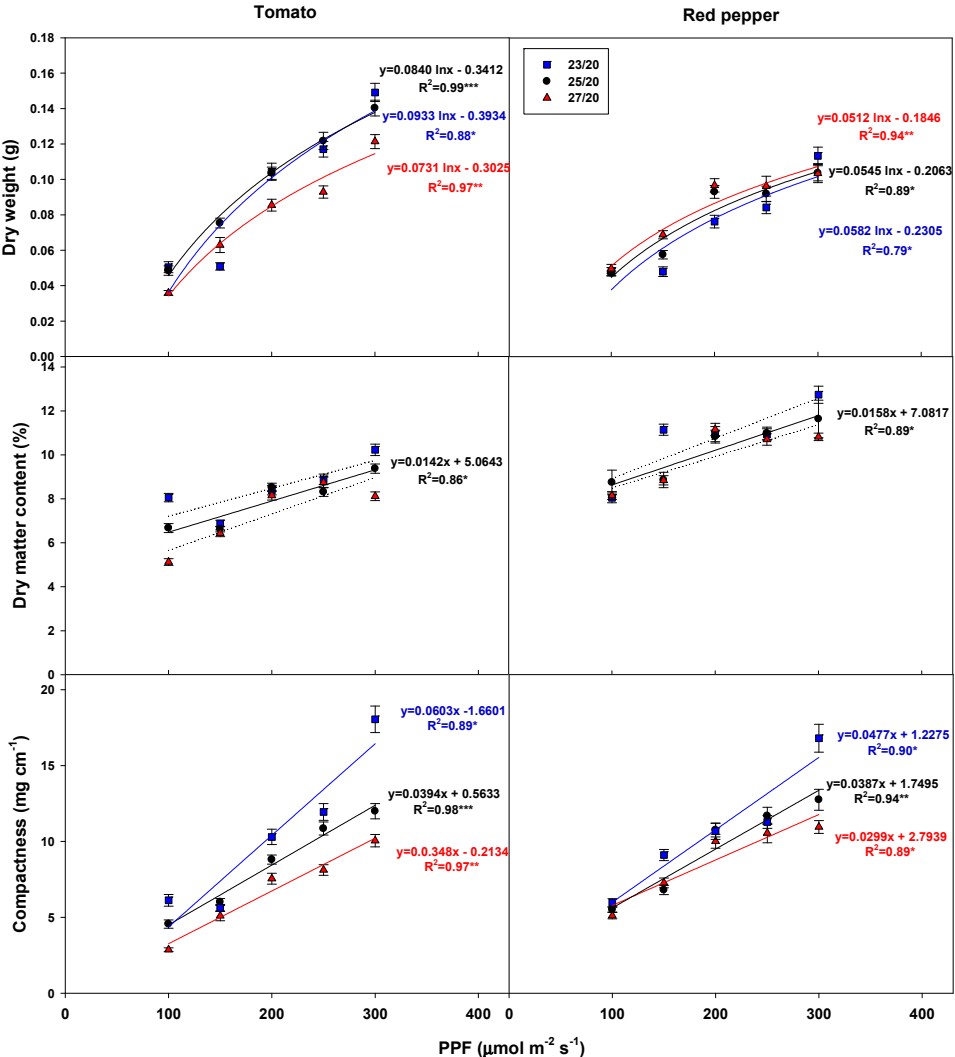

**Figure 10.** Regression analyses of the dry weight, dry matter content, and compactness related to the light intensity and photoperiod. The value of each point is the average dry weight, dry matter content, and compactness of tomato at 17 days after sowing (DAS) and red pepper at 25 DAS, as influenced by the different light intensities and air temperatures of the photo- and dark periods. Bar means the standard error. The regression equation and $R^2$ coefficients are presented when statistically significant (solid line) but not when nonsignificant (dotted line) *, **, and *** indicate significance at $p < 0.05$, 0.01, and 0.001, respectively.

Unlike DW, the DMC and compactness at each temperature displayed linear correlations instead of logarithmic correlations to PPF in both cultivars. In the case of DMC, only data at 25/20 °C showed a significant relationship ($p < 0.05$) with PPF. For compactness, however, strong and significant linear correlations with PPF (coefficients of determination were 0.89 and above) could be found at all temperature settings in both cultivars. In tomato, the slope of the regression line between PPF and compactness at 23/20 °C was the highest, while that at 27 °C was similar to that at 25 °C, although with a lower altitude. In the red pepper, the slope of the regression line between PPF and compactness decreased as the temperature increased. All the slopes in both cultivars were positive, showing that the increase in PPF all led to an increase in compactness at all temperatures, albeit with different magnitudes.

### 3.2.4. Light Use Efficiency

In tomato, the LUE at 27/20 °C was generally lower than that at 25/20 and 23/20 °C (Figure 11). Under the same temperature, there was no clear difference in LUE among treatments with different PPFs. In the case of red pepper, no clear trend in LUE with regard to PPF or temperature could be observed. However, the LUE of the 100/23 treatment (1.11 g·mol$^{-1}$·m$^{-2}$·d$^{-1}$) was significantly higher than that of all other treatments.

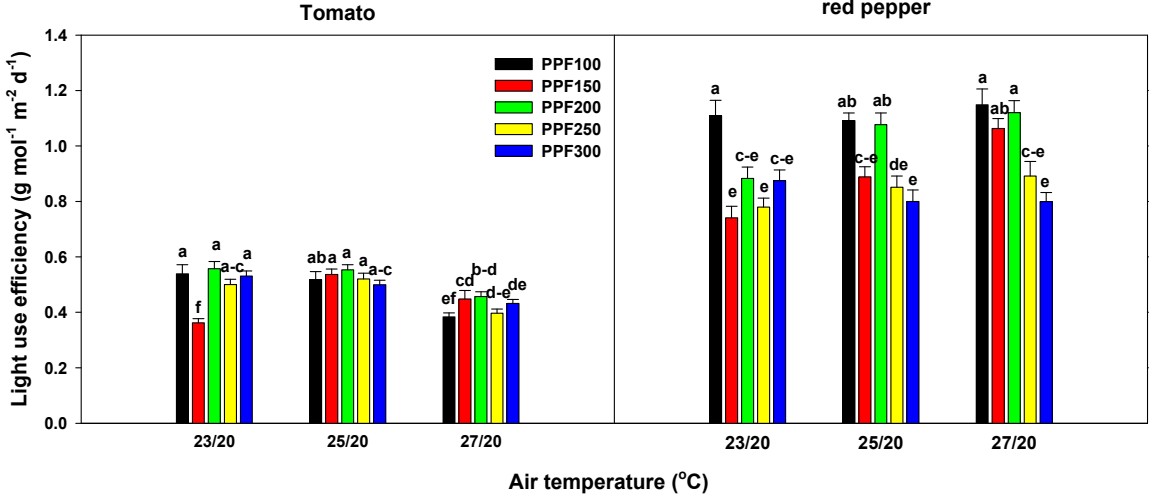

**Figure 11.** Light use efficiency (LUE) of tomato at 17 days after sowing (DAS) and red pepper at 25 DAS, as influenced by the different light intensities and air temperatures of photo- and dark periods (n = 7, r = 3).

### 3.3. Effects of Different Air Temperatures (Photo and Dark Periods) and Light Intensities on Tomato and Red Pepper Seedlings

The first flower nodes of tomato and red pepper under different air temperatures and PPFs were 9.0–10.5 and 7.9–10.3, respectively (Table 3). The first flower node of tomato and red pepper in the 27/20 °C treatment was the lowest, while PPF did not show a significant difference (tomato) or only a small effect (red pepper). In red pepper, both a high light intensity and air temperature accelerated flowering.

## 4. Discussion

### 4.1. Effects of Photoperiod and Light Intensity on Tomato and Red Pepper Seedlings

Increasing the DLI from 4.32 to 21.60 mol·m$^{-2}$·d$^{-1}$ influenced the shoot length and stem diameter of tomato and red pepper seedlings differently. The shoot length of tomato and red pepper was the highest at 200 µmol·m$^{-2}$·s$^{-1}$ (DLI 8.64 for 12 h·d$^{-1}$, 11.52 for 16 h·d$^{-1}$, and 14.40 mol·m$^{-2}$·d$^{-1}$ for 20 h·d$^{-1}$) and decreased under higher PPF treatments (Figure 4). This result agrees with a previous study,

in which the hypocotyl length of lettuce decreased as the DLI increased from 5.8 to 25.9 mol·m$^{-2}$·d$^{-1}$. The hypocotyls of lettuce at a DLI of 8.6 mol·m$^{-2}$·d$^{-1}$, a lower PPF, and a higher photoperiod (24 h·d$^{-1}$) were higher than those grown under a higher PPF and lower photoperiod (16 h·d$^{-1}$) [14]. Scion and rootstock seedlings benefit from an extended hypocotyl length, as this increases the grafting success rate and reduces the rooting from the scion after transplant [15]. However, elongated hypocotyls are not beneficial for nongrafted seedlings because this property may lead to weak transplants [16].

An increase in PPF promoted the stem diameter of tomato and red pepper seedlings among all the tested photoperiods. The stem diameters of tomato and red pepper were the highest at 250 (DLI 10.80 for 12 h·d$^{-1}$, 14.40 for 16 h·d$^{-1}$, and 18.00 mol·m$^{-2}$·d$^{-1}$ for 20 h·d$^{-1}$) and 200 μmol·m$^{-2}$·s$^{-1}$ (DLI 8.64 for 12 h·d$^{-1}$, 11.52 for 16 h·d$^{-1}$, and 14.40 mol·m$^{-2}$·d$^{-1}$ for 20 h·d$^{-1}$), respectively (Figure 4). These results agree with a previous study, as increasing the DLI from 6.1 to 11.8 mol·m$^{-2}$·d$^{-1}$ increased the stem diameter of cucumber and pepper seedlings [17]. Based on the results of this study, we can present the optimal range for the stem growth of tomatoes and red peppers.

Increasing the DLI from 4.32 to 21.60 mol·m$^{-2}$·d$^{-1}$ promoted the leaf characteristics—including leaf number, leaf area, and leaf area index (LAI)—of both cultivars. The leaf number of tomato and red pepper was the greatest at 250 μmol·m$^{-2}$·s$^{-1}$, while there were no significant differences if the PPF was higher than 200 μmol·m$^{-2}$·s$^{-1}$. Leaf area and LAI showed a similar pattern of leaf number results. The leaf area and LAI of tomato and red pepper were the greatest among all the tested photoperiods at 250 and 200 μmol·m$^{-2}$·s$^{-1}$, respectively (Figure 5). In this study, DLIs of 18.00 and 14.40 mol·m$^{-2}$·d$^{-1}$ were sufficiently high in at least tomato and red pepper seedlings, in which individual leaves expanded in response to a given cultivation area. Plants grown in a high light intensity generally have thick leaves with a low specific leaf area [18], partly due to the added layers of palisade or extended palisade cells [19]. This increases the number of chlorophyll and photosynthesis-related enzymes, improving the photosynthesis rate of the plant [20]. However, the photosynthesis rate increase obtained after the plant leaves adapted to a high light intensity reduced the efficiency of light capture per unit biomass at lower intensities [21]. Leaf area was decreased with photoperiod from 12 to 21 h·d$^{-1}$ in sweet potato [22]. Leaf area was smaller under continuous light (24 h·d$^{-1}$) than plants grown under 20 h·d$^{-1}$, which resulted in the better light capture of cucumber [17]. A reduction in leaf area as an effect of a longer photoperiod has been reported in Eustoma [23] and tomato [24]. Previous studies have reported that the leaf area decreased while the leaf area index (LAI) increased as the plant density increased in pepper [25,26]. Higher plant density conditions (higher LAI) increased the light capture capacity, and, as a result, the growth of sweet pepper increased [27]. Additionally, increases in LAI above a threshold (from approximately 3.5 to 4 for most plants) do not increase the radiation interference [28]. This explains why increases in plant growth above an LAI threshold occur at a given space of seedlings.

Under a given temperature of 25/20 °C, the dry weight (DW), dry matter content (DMC), and compactness of tomato and red pepper seedlings showed a positive relationship with DLI as the DLI increased from 4.32 to 21.60 mol·m$^{-2}$·d$^{-1}$ (Figure 6). Prolonged photoperiod is a widely used technique in greenhouse and closed systems and has provided increased growth and yield for many species, such as lettuce [14,29], cucumber [30], tomato, and red pepper [17]. Plant growth by an increase in PPF and photoperiod is an almost linear relationship for many plant species under a controlled environment [14,17,31]. During vegetative growth, tomato seedlings positively respond to the increased DLI at the leaf level via increased PPF [32,33] and photoperiod [24,34]. Further study will have to be carried out later with a greater DLI (over 21.6 mol·m$^{-2}$·d$^{-1}$) than this study to find the maximum point. In this study, the same DLI was derived from three points (8.64, 10.80, and 14.40 mol·m$^{-2}$·d$^{-1}$) as a different combination of PPF and photoperiod. While many previous studies have shown that increasing the DLI increases the dry mass of many plants, including Achemenes [35,36], lettuce [14,37–39], and radish [37], few studies have described how different combinations of PPF and photoperiod in the same DLI interact [8,40]. The DW of tomato was increased at a relatively low PPF and long photoperiod more than at a higher PPF and shorter photoperiod. In the case of red pepper, the same tendency was shown, except for DLI 8.64. The compactness of tomato and red pepper

also increased at a relatively low PPF and long photoperiod more than at a higher PPF and shorter photoperiod. Kelly et al. [31] reported a similar pattern of these results in lettuce, which explained as photosynthesis efficiency. At a higher PPF above a threshold, the photosynthesis efficiency decreased with increasing PPF, where further PPF increases did not increase photosynthesis. After reaching the light saturation point, only a photoperiod increase will enhance the net photosynthesis. This may explain why using a lower PPF with a longer photoperiod, under the same DLI conditions, is more effective than using a higher PPF with a shorter photoperiod [41]. Interestingly, the growth of red pepper under 20/100 and 20/150 was higher than that under 16/300. Even in higher DLI conditions, reduced growth due to higher PPF proves that excessive light energy has been a limiting factor. Furthermore, reducing the PPF can reduce the number of lamps, thereby reducing the initial installation cost of the system [5].

*4.2. Effects of Different Air Temperatures (Photo and Dark Periods) and Light Intensities on Tomato and Red Pepper Seedlings*

The response to changes in the PPF of tomato and red pepper in terms of shoot length differed significantly depending on the photo/dark period air temperature (Figure 8). At 23/20 °C, tomato and red pepper generally showed increasing and decreasing patterns, with shoot length increasing as the PPF increased up to 200 $\mu$mol·m$^{-2}$·s$^{-1}$ in tomato and 250 $\mu$mol·m$^{-2}$·s$^{-1}$ in red pepper and then decreasing as the PPF increased further to 300 $\mu$mol·m$^{-2}$·s$^{-1}$. Many studies of the relationship between light intensity and shoot length have been reported in relation to gibberellic acids (GAs) [42–44]. The effect of low PPF on shoot elongation was due to an increase in GA$_1$ biosynthesis and a decrease in GA$_1$ catabolism [45]. The epicotyl length of brassica seedlings increased from 50 to 250 $\mu$mol·m$^{-2}$·s$^{-1}$ and decreased further to 500 $\mu$mol·m$^{-2}$·s$^{-1}$, while the hypocotyl length decreased as the PPF increased from 50 to 500 $\mu$mol·m$^{-2}$·s$^{-1}$ [46]. Lower endogenous GAs were observed under a higher PPF. Nevertheless, the response of shoot length to an increase in temperature at the same PPF was consistent in each species. For tomato, the shoot length at the same PPF was higher at 25/20 °C than at 23/20 °C, but there was no further significant improvement as the temperature increased to 27/20 °C. Red pepper, on the other hand, still showed a significant increase in shoot length as the photoperiod temperature increased up to 27 °C in all PPFs. Plants can adjust their shoot length when irradiated with strong light above their photosynthetic saturation point [47]. Previous research in sweet pepper showed that the maximum height was 24/21 or 28/15 °C under sunlight in a greenhouse [10]. In this study, under a sufficiently long photoperiod, strong light shortened the length of tomatoes.

The response of stem diameter to changes in the PPF and temperature was more straightforward. In both species, stem diameter showed an increasing trend as PPF increased at the same temperature, although there was no significant difference in red pepper among PPFs higher than 150 $\mu$mol·m$^{-2}$·s$^{-1}$ at 25 and 27 °C. Across different temperatures at the same PPF level, however, the stem diameter in tomato showed a slight decrease as the temperature increased from 25 to 27 °C, while that of red pepper was not significantly different between 25 and 27 °C, except at 150 $\mu$mol·m$^{-2}$·s$^{-1}$. Under sufficient light conditions, the optimal daytime air temperature of the tomato is estimated to be 23 °C, and in the case of red peppers, it is considered to be higher than that of tomato. Air temperatures above the optimal range obstruct cell elongation and differentiation [48].

In tomato, while there was no clear pattern in leaf number in response to changes in temperature and PPF, leaf area and LAI displayed a clear upward trend as PPF increased under all temperature settings. When comparing temperatures under the same PPF, however, there was a decrease in leaf number as the temperature increased from 25 to 27 °C, except at a PPF of 300 $\mu$mol·m$^{-2}$·s$^{-1}$. In red pepper, leaf number showed an overall increase as the PPF increased under all air temperature settings (Figure 9). Leaf characteristics including leaf number and leaf area are closely related to the photosynthesis (especially light harvest) of plants. Planting space and planted area are crucial factors that greatly affect leaf characteristics. In a limited space, after securing a favorable area for photosynthesis it will no longer increase [49].

An increase in PPF led to a higher DW in both species, but at higher PPFs this effect was less pronounced than at low PPFs. More importantly, the models revealed the differences between tomato and red pepper in the response to changes in PPF under different temperatures (Figure 10). At photoperiod temperatures of 23 and 25 °C, tomato showed similar correlation curves between DW and PPF. However, at the higher temperature of 27 °C, while the slope of the curve at low PPFs was similar to that of the curves at lower temperatures, the slope at high PPFs was flatter. On the other hand, red pepper displayed the same correlation curve between DW and PPF at all tested temperatures, albeit the curve at 27 °C was slightly higher at altitude than that at 25 °C, which in turn was higher than that at 23 °C, showing that red pepper plants accumulated more DW at higher temperatures. Unlike DW, the DMC and compactness at each temperature displayed linear correlations instead of logarithmic correlations to PPF in both cultivars. For compactness, strong and significant linear correlations with PPF could be found at all temperature settings in both cultivars. In tomato, the slope of the regression line between PPF and compactness at 23/20 °C was the highest, while that at 27 °C was similar to that at 25 °C, although with a lower altitude. In red pepper, the slope of the regression line between PPF and compactness decreased as the temperature increased. All the slopes in both cultivars were positive, showing that the increase in PPF led to an increase in compactness at all temperatures, albeit with different magnitudes. The optimum temperature for the plant growth of both cultivars that prefer high lighting requirements was different. A previous study recommended that optimum tomato growth occurs at air temperatures from 17 to 23 °C and stops at a maximum of 33 °C and a minimum of 12 °C [50]. The optimal temperature for pepper seedling development was reported to be approximately 25 °C [51], and it reaches at a minimum of 14.5 °C [52]. In tomato under high-temperature conditions, a higher PPF was a limiting factor in photosynthesis. At high temperatures, the photosystem II activity was reduced in tomato plants [11,53]. The compactness of both cultivars at 23/20 °C was greater than those at the other air temperature treatments. Generally, high DW, DMC, and compactness are defined by high-quality seedling standards [14]. Ohyama et al. [54] reported that, by increasing the air temperature from 20 to 28 °C under continuous light conditions (24 h·d$^{-1}$), the highest compactness was observed under the 20 °C treatment.

Increasing the PPF from 100 to 300 μmol·m$^{-2}$·s$^{-1}$ differently influenced the LUE by the photo- and dark-period air temperatures of tomato and red pepper seedlings (Figure 11). The LUE of tomato at 200 μmol·m$^{-2}$·s$^{-1}$ was the highest among treatments, with values of 0.56 (23/20 °C), 0.55 (25/20 °C), and 0.46 mg·mol$^{-1}$·m$^{-2}$·d$^{-1}$ (27/20 °C). In the case of red pepper, the LUE values under 25/20 and 27/20 °C at 200 μmol·m$^{-2}$·s$^{-1}$ were 1.07 and 1.11 mg·mol$^{-1}$·m$^{-2}$·d$^{-1}$, respectively. Generally, mature tomato and red pepper are considered high light-requiring crops compared to other plants, and a DLI of 20–35 mol·m$^{-2}$·d$^{-1}$ is recommended for their optimal growth and production [10,50,55,56]. The LUE increases with an increase in the light energy received by leaves. This can be enhanced by well-designed light conditions, a reduction in distance between lamps and plants, and an increase in the planting density [57]. Normally, the LUE of tomato seedlings increases linearly from zero up to unity with increasing the LAI from 0 to 3 [58].

The first flowering nodes of tomato and red pepper under different air temperatures and PPFs were 9.0–10.5 and 7.9–10.3, respectively (Table 3). The first flowering node of tomato and red pepper in the 27/20 °C treatment was the lowest, while PPF did not show a significant difference (tomato) or only a small effect (red pepper). Sato et al. [59] found that mean daily temperatures over 25 °C reduced the fruit and seed set in tomato. The flower development of tomato was accelerated as the average daily air temperature increased from 19 to 24 °C [60]. Fruit set is reduced in tomato when their average maximum day temperatures are above 30 °C and night temperatures are above 20 °C during anthesis [50]. Low temperatures and irradiation cause improper ovary development, unviable pollen, the malformation of flowers, fruit puffiness, and blotchy ripening [61]. For this reason, some research has recommended that the optimum daytime air temperature of red pepper is approximately 23 °C [62].

**Table 3.** First flowering node of tomato and red pepper as affected by the different light intensities and air temperatures of photo- and dark periods at 44 and 52 days after sowing, respectively (n = 7, R = 3). Different letters indicate significant differences by Duncan's multiple range test at *p* = 0.05.

| Air Temperature (°C) | Light Intensity ($\mu mol \cdot m^{-2} \cdot s^{-1}$) | Node of First Flower | | | |
|---|---|---|---|---|---|
| | | Tomato | | Red Pepper | |
| 23/20 | 100 | 9.9 | bc | 9.6 | b |
| | 150 | 10.5 | a | 9.2 | bc |
| | 200 | 10.1 | ab | 8.9 | cd |
| | 250 | 10.2 | ab | 9.0 | cd |
| | 300 | 10.0 | cb | 8.5 | d–g |
| 25/20 | 100 | 9.7 | c–e | 10.3 | a |
| | 150 | 9.3 | ef | 8.8 | cd |
| | 200 | 9.3 | ef | 8.6 | d–g |
| | 250 | 9.4 | d–f | 8.8 | c–e |
| | 300 | 9.8 | b–d | 8.3 | e–h |
| 27/20 | 100 | 9.2 | f | 8.7 | e–f |
| | 150 | 9.0 | f | 8.1 | gh |
| | 200 | 9.2 | f | 8.2 | f–h |
| | 250 | 9.3 | ef | 7.9 | h |
| | 300 | 9.6 | c–e | 8.0 | gh |
| Significance | | | | | |
| Air temperature (A) | | *** | | *** | |
| Light intensity (B) | | NS | | *** | |
| Interaction (A × B) | | *** | | *** | |

*** indicate significance at *p* < 0.001

## 5. Conclusions

The results of this study highlight the differences in nature between tomato and pepper, as well as the complex interactions among PPF, photoperiod, and temperature on the growth and morphology of plants. The dry matter content and compactness displayed linear relationships with DLI and PPF, while dry weight showed trends of saturating curves with these factors. The trends of many parameters, most notably dry weight and compactness, in response to PPF could be shifted by the change in photoperiod or air temperature, and vice versa. Tomato exhibited a preference for a temperature of 23/20 or 25/20 °C, especially in terms of biomass gain and light use efficiency, while pepper show no change in these aspects under the temperature range from 23/20 to 27/20 °C. There is great potential in applying these results to control the quality of plants and improve the efficiency in the cultivation of tomato and pepper in CTPS.

**Author Contributions:** Conceptualization, methodology, data curation, formal analysis, investigation, writing, H.H.; conceptualization, methodology, writing, funding acquisition, S.A.; writing, M.D.P.; review and editing, M.C.; conceptualization, supervision, validation, review and editing, funding acquisition, C.C. All authors have read and agreed to the published version of the manuscript.

**Funding:** This research was funded by the Rural Development Administration (PJ013840), "Development of a plant factory type seedling production system to produce standard fruit vegetable seedlings linked with a grafting robot".

**Conflicts of Interest:** The authors declare no conflict of interest.

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
