# Peer review of "The Combined Conditions of Photoperiod, Light Intensity, and Air Temperature Control the Growth and Development of Tomato and Red Pepper Seedlings in a Closed Transplant Production System"

_sustainability, doi:10.3390/su12239939_

Round 1

Reviewer 1 Report

In this article, the authors examine the combined effects of photoperiod, light intensity, and air temperature on the growth of tomato and pepper.  The results of this study were clearly presented, while the discussion is very well written. I provide below a few suggestions that, if the authors decide to implement into the paper, the paper will be improved.

Comments

Line 4 (Title): The word young should be removed from the title of the article since seedlings refer to young plants.

Lin 23 (Abstract): the term DIFs should be analyzed since is mentioned at first time in the text.  

Lines 81-88. The date of sowing should be added.

Line 180 and 191: Photoperiod unit should be corrected.

Lines 309-309: The data presented in tables 3 and 4 should be presented in a new table 3.

Figure 4 and 5: the font size should be larger especially for letters that shows the significance differences between the treatments.

Author Response

Response to Reviewer 1 Comments

Point 1: Line 4 (Title): The word young should be removed from the title of the article since seedlings refer to young plants.

Response 1: We removed the word “young”

Point 2: Line 23 (Abstract): the term DIFs should be analyzed since is mentioned at first time in the text. 

Response 2: We added the term “DIF”, DIFs (difference between the day and night temperature).

Point 3: Lines 81-88. The date of sowing should be added.

Response 3: We added the date of sowing, “On 1 February”

Point 4: Line 180 and 191: Photoperiod unit should be corrected.

Response 4: We corrected the photoperiod unit, h‧d-1

Point 5: Lines 309-309: The data presented in tables 3 and 4 should be presented in a new table 3.

Response 5: We made a new table 3.

Point 6: Figure 4 and 5: the font size should be larger especially for letters that show the significance differences between the treatments.

Response 6: We enlarged the font size (11) of letters that shows the significance differences

Reviewer 2 Report

This paper investigates the combined effect of photoperiod, light intensity and temperature on tomato and pepper seedlings. The experiment consists of two main parts, both used 15-15 treatments; this combination is capable of showing some tendencies regarding the interconnections of the targeted factors. The experiment is well designed; however, the methodology is not extensively documented.

General comments

Methodology regarding flowering data is totally missing from the MS. Also, the dimension is missing from the tables and from the text as well. Please elaborate.

The discussion part is sometimes very hard to understand due to its unclear and burdensome wording. Shortening the sentences might help the better understanding.

Detailed comments

line23: DIF: undefined abbreviation, please define.

line 41: change many to numerous

line 45: increase production curve

line 51: the most important ones

line 55: how plants respond

line 56: please reword this sentence

line 64: focus only on ...

line 163: within the same photoperiod

line 34 : at least in the case of tomato and red pepper seedlings

line 342: partly due to extra layers

line 377: Sentence starting with: This may explain … Please reword.

line 391: abbreviation GA: please elaborate.

line 464: It is irrelevant to speak about fruit set, when it was obviously not measured in this experiment. I recommend changing this part and use sources that deal exactly with flower development, especially with the time of the development of the first flower. At the same time, seedlings often develop flowers earlier due to unfavorable environmental conditions, therefore it is rather a two-sided, equivocal trait, that should be mentioned in the MS.

Author Response

Response to Reviewer 2 Comments

Point 1: line23: DIF: undefined abbreviation, please define.

Response 1: We added the abbreviation of “DIF”, DIFs (difference between the day and night temperature).

Point 2: line 41: change many to numerous

Response 2: We changed the word “many to numerous”

Point 3: line 45: increase production curve

Response 3: We changed the word “increase the production rate” to “increase production curve”

Point 4: line 51: the most important ones

Response 4: We added the word “ones”

Point 5: line 55: how plants respond

Response 5: We removed the word “the”

Point 6: line 56: please reword this sentence

Response 6: We changed the sentence to “There have been many studies on the effects of PPF, photoperiod, and air temperature as separate factors affecting plant growth in closed systems such as CTPS”

Point 7: line 64: focus only on ...

Response 7: We changed the word “only focus on” to “focus only on”

Point 8: line 163: within the same photoperiod

Response 8: We changed the word “with to within”

Point 9: line 234 : at least in the case of tomato and red pepper seedlings

Response 9: We changed the word to “at least in the case of red pepper seedlings”

Point 10: line 342: partly due to extra layers

Response 10: We changed the word “due in part to extra layers” to “partly due to extra layers”

Point 11: line 377: Sentence starting with: This may explain … Please reword.

Response 11: We changed the sentence to “This may explain why using a lower PPF with a longer photoperiod, under the same DLI conditions, is more effective than using a higher PPF with a shorter photoperiod”

Point 12: abbreviation GA: please elaborate.

Response 12: We added the abbreviation of GAs, “gibberellic acids (GAs)”.

Point 13: line 464: It is irrelevant to speak about fruit set, when it was obviously not measured in this experiment. I recommend changing this part and use sources that deal exactly with flower development, especially with the time of the development of the first flower. At the same time, seedlings often develop flowers earlier due to unfavorable environmental conditions, therefore it is rather a two-sided, equivocal trait, that should be mentioned in the MS.

Response 13: We added the reference about flower developmental changes by air temperature.
